# Classification of Neurological Patients to Identify Fallers Based on Spatial-Temporal Gait Characteristics Measured by a Wearable Device

**DOI:** 10.3390/s20154098

**Published:** 2020-07-23

**Authors:** Yuhan Zhou, Rana Zia Ur Rehman, Clint Hansen, Walter Maetzler, Silvia Del Din, Lynn Rochester, Tibor Hortobágyi, Claudine J. C. Lamoth

**Affiliations:** 1Department of Human Movement Sciences, University Medical Center Groningen, University of Groningen, 9713 AV Groningen, The Netherlands; t.hortobagyi@umcg.nl; 2Translational and Clinical Research Institute, Faculty of Medical Sciences, Newcastle University, Newcastle Upon Tyne NE4 5PL, UK; Rana.zia-ur-Rehman@newcastle.ac.uk (R.Z.U.R.); silvia.del-din@newcastle.ac.uk (S.D.D.); lynn.rochester@newcastle.ac.uk (L.R.); 3Department of Neurology, University Hospital Schleswig-Holstein, Campus Kiel, 24105 Kiel, Germany; c.hansen@neurologie.uni-kiel.de (C.H.); w.maetzler@neurologie.uni-kiel.de (W.M.); 4Newcastle Upon Tyne Hospitals NHS Foundation Trust, Newcastle Upon Tyne NE7 7DN, UK

**Keywords:** gait analysis, machine learning, inertial measurement units, neurological disorders, falls

## Abstract

Neurological patients can have severe gait impairments that contribute to fall risks. Predicting falls from gait abnormalities could aid clinicians and patients mitigate fall risk. The aim of this study was to predict fall status from spatial-temporal gait characteristics measured by a wearable device in a heterogeneous population of neurological patients. Participants (*n* = 384, age 49–80 s) were recruited from a neurology ward of a University hospital. They walked 20 m at a comfortable speed (single task: ST) and while performing a dual task with a motor component (DT1) and a dual task with a cognitive component (DT2). Twenty-seven spatial-temporal gait variables were measured with wearable sensors placed at the lower back and both ankles. Partial least square discriminant analysis (PLS-DA) was then applied to classify fallers and non-fallers. The PLS-DA classification model performed well for all three gait tasks (ST, DT1, and DT2) with an evaluation of classification performance Area under the receiver operating characteristic Curve (AUC) of 0.7, 0.6 and 0.7, respectively. Fallers differed from non-fallers in their specific gait patterns. Results from this study improve our understanding of how falls risk-related gait impairments in neurological patients could aid the design of tailored fall-prevention interventions.

## 1. Introduction

Falls are one of the leading causes of mortality, morbidity, and make up a substantial portion of health care costs worldwide [1]. Falls have a multifactorial origin and usually involve multiple interrelated intrinsic, as well as extrinsic factors [2]. There has been a plentitude of reviews aimed at the epidemiology of fall risk in the aging population. Identified risk factors in the aging population are reduced lower extremity strength, sarcopenia, dizziness, vision impairments, a decline in cognitive function, higher prevalence of comorbidities, polypharmacy, DBI drug use (cumulative anticholinergic and sedative exposure), depression, and extrinsic factors, such as poor lighting in the house, loose rugs, and slippery surface [2,3,4,5]. Healthy people can adapt easily to environmental perturbations, such as recovering from slipping or tripping or walking on uneven surfaces. With aging and/or age-related pathology, the ability to adapt to environmental perturbation while walking is diminished. Yet, the most consistent predictors of falls are impairments gait and balance disorders [6,7]. Moreover, various neurological disorders further increase the risk of falls by deteriorating specific nervous system functions contributing to gait and balance [8]. Therefore, the incidence of falls is high in neurological patients; compared with healthy subjects, neurological patients had a 49% increased risk of falling within 20 months [9,10]. Patients with Parkinson’s Disease have falls at least once during their disease journey [11].Therefore, detecting gait impairments by wearable devices, such as inertial measurement units (IMUs), during walking could help clinicians identify patients prone to falling [12]. IMUs are attractive alternatives to laboratory motion analysis systems due to their small size, light weight, portability, low cost, and their simple use in the real world.

Spatial-temporal gait variables derived from IMU recordings are outcome parameters for the prediction of falls in patient groups. Fallers compared with non-fallers revealed higher standard deviations and coefficients of variation of stride time, swing time, stance time, and percentage stance time [13]. There exists a strong inter-relationship between cognition and gait control, since gait and cognitive function share cortical areas and several neurotransmitters [14]. An additional cognitive task while walking in clinical populations or older adults, might stress the system by competing for cortical resources, placing the patient at an increased fall risk while performing dual tasks [15,16]. Therefore, the classification accuracy of fallers and non-fallers might be more accurate when walking while performing a dual task. In a previous study, spatial-temporal gait variables obtained during dual task increased classification performance of fallers and non-fallers in 377 older adults compared to single task walking from 0.57 to 0.67 as quantified by the Area Under the receiver operating Curve (AUC) [17].

Different computational approaches, such as supervised machine learning approaches artificial neural network (ANN), K-nearest neighbors (KNN) or support vector machine (SVM), have become popular for the classification of fall risk using different types of tests and/or activities performed by healthy older persons or distinct group of patients [18,19]. Based on time-frequency domain features, different activities were classified using ANN, KNN, quadratic support vector machine (QSVM), and ensemble bagged tree (EBT), the classification accuracy of 85.8%, 91.8%, 96.1%, and 97.7% was obtained for fall detection, respectively [18]. Using a functional movement test, including walking and sit-to-stand with data from foot force sensors, different KNN-based classifiers were compared by classification accuracy of falls for older adults [19]. Although accuracy was reported to be 100% for local mean pseudo nearest neighbor method, the number of subjects included was small and was relatively healthy without any neurological or orthopedic condition that would affect their gait pattern.

Although these studies have successfully classified falls only for general older populations, or just focused on one specific neurologic disorder, the identification of gait impairments for the classification of falls in a more heterogeneous neurological population based on spatial-temporal variables requires a different approach [20,21].

Since the deconstruction of gait into clinically observable spatial-temporal components, such as shorter steps or longer strides, could assist clinicians in having a gait assessment for falls under direct clinical observation, there are advantages using these gait variables derived from IMUs to classify falls. For example, computational approaches can perform automated analyses of multivariate datasets and can deal with interdependency (collinearity) among gait variables from IMU, such as walking speed, mean stride times, and variability in stride times. However, many of these popular computational methods in previous studies with a relatively small sample size will result in overfitting due to the structure of spatial-temporal gait data. Alternatively, multivariate partial least square (PLS) regression or discriminant analysis (DA) analysis can be applied. PLS is a technique that combines features from principal component analysis and multiple regression and is not impeded by collinearity among variables. Besides, partial least square discriminant analysis (PLS-DA) is suitable for gait data in which classes (faller vs. non-fallers) are predicted from a relatively large set of independent (gait) variables with relatively few observations [22]. Clinically, the results of such a computational approach can assist clinicians in interpreting gait performance of patients and use it as a prevention tool that can identify patients with high fall risk. By extracting those gait features that include the information to distinguish classes, tailored intervention programs to reduce the probability of future falls can be developed [23].

The aim of the present study was to establish a quantitative model to classify fallers and non-fallers using spatial-temporal gait characteristics and to identify the specific gait characteristics that contributed to the classification model to target to mitigate falls risk. Considering the strong interrelation of many spatial-temporal gait variables, we hypothesized that, based on a subset of general gait features, fallers can be classified from non-fallers even in a heterogeneous group of neurological patients, and the classification accuracy will be improved while walking with an additional dual task.

## 2. Materials and Methods

### 2.1. Participants

Participants (*n* = 384, age range: 49–80 years) with neurological disorders were recruited from three neurology wards of the University Hospital of Tubingen between September 2014 and April 2015 [24]. The distribution of the major neurological disorders was: 19% Parkinson’s disease (8% of fallers), 19% stroke (5% of fallers), 11% epilepsy (4% of fallers), 10% pain syndromes (3% of fallers), 9% multiple sclerosis (4% of fallers), 7% central nervous system tumor (2% of fallers), 6% vertigo (2% of fallers), 6% dementia (2% of fallers), and 6% meningitis/encephalitis (1% of fallers) (see Reference [24] for demographics). Participants were included if they were able to walk 20 m with or without walking aid. Exclusion criteria were: inability to give informed consent, a falling frequency of more than one fall per week, and impaired cognition (Mini-Mental State Examination (MMSE) score ≤10 [24]). Participants were classified as fallers if they had fallen at least once during a two-year period before recruitment. The ethics committee of the medical faculty of the University of Tübingen approved the study (No. 356/2014BO2), and all participants gave written informed consent prior to participation. The investigation was carried out following the rules of the Declaration of Helsinki of 1975, revised in 2013.

### 2.2. Procedure

Participants were instructed to walk 20 m at a comfortable speed (Single Task; ST), with a dual task (DT) containing mainly a motor component (walking and checking boxes on a paper sheet, DT1) and with a cognitive task (serial 7 s subtraction, DT2) [25]. A complete gait dataset was available for 349 of the 384 participants for ST, wherein 274 participants performed DT1 and 306 participants performed DT2. Table 1 shows participants’ demographics for the two groups.

### 2.3. Data Collection

An IMU-based wearable sensor system was attached with straps around the middle part of the foot around the shoe, to collect data during walking. The IMU system had 3D accelerometers (±8 g), 3D gyroscopes (±2000°/s), and 3D magnetometers (±1.3 Gs), resulting in nine degrees of freedom (Rehawatch, Hasomed, Magdeburg, Germany) [26].

In each walking task, the following 27 gait variables were extracted from the accelerometer signals: mean and standard deviation of stride duration (s), stride length (m), stride velocity (m/s), number of steps (*n*), percent of stance (%), stance time (s), percent of swing (%), swing time (s), symmetry stance phase, symmetry swing phases, single support phase (s), ankle dorsiflexion at heel strike (°), plantar flexion at toe-off (°), circumduction of gait (cm), percent of gait cycle time variability (%), and percent of gait cycle spatial variability (%) [26]. In certain combinations, these variables are sensitive to aging and neurodegenerative diseases [10,12,27].

### 2.4. Statistical Analysis

The data set obtained from ST, DT1, and DT2 were analyzed separately. First, we evaluated if the 27 gait variables correlated linearly or non-linearly with each other, determining the choice of the subsequently used classification method. Partial Least Square Discriminate Analysis (PLS-DA) was applied to classify fallers from non-fallers.

The 27 gait outcomes were the independent variables, and the dependent variable was the classification of participants as non-fallers (Class 0) and fallers (Class 1). The PLS-DA model results in a data dimensionality reduction (Latent Variables; LVs) and provides parameters to evaluate the quality of the prediction and classification of the model. Besides, the Variable of Importance (VIP) gives information about the contribution of individual gait variables to the model. The VIP is calculated as:(1)$$VIP_{j}=\sqrt{p\sum_{k=1}^{N}\frac{\left[SS_{k}\left(\frac{w_{kj}}{{\left|\left|w_{k}\right|\right|}^{2}}\right)\right]}{\sum_{k=1}^{N}{\left(SS\right)}_{k}}}.$$

p is the total number of gait variables in the model. N is the number of LVs in the PLS-DA model, k represents the exact component of LV, $ss_{k}$ explains the sum of the variance of the LVs, $w_{kj}\;$ quantifies the contribution of variable j  according to the kth LV, and $w_{k}$ is the contribution of the kth LV.

Variables with a VIP >1 indicate a significant contribution of the variable to the classification model. The non-standardized data were tested for normality with the Shapiro-Wilk normality test in R. Since not all variables were normal distributed, a non-parametric Mann–Whitney–Wilcoxon test in R programming was applied [28] to test if the VIP variables >1 were significantly different between non-fallers and fallers.

The standard of the goodness of the model was addressed by the PLS-DA model parameters Q^2^, R^2^X and R^2^Y. The values of these parameters need to be higher than zero in order to have an acceptable classification model. To avoid a too complex model with poor predictability (the problem of overfitting), leave one out cross-validation method was used with PLS-DA classification for assessing the classification model building [29].

Classification performance evaluation of the PLS-DA model was assessed by the receiver operating characteristic curve (ROC curve) and the Area Under the Curve (AUC). In addition, for the non-fallers group and fallers group, their corresponding true positive rate (sensitivity) and true negative rate (specificity) were calculated based on the confusion matrix.

## 3. Results

### 3.1. Classification of Fallers during Single and Dual Task Walking

The first five LVs for ST explained 61.4% variance of the original gait variables. Additional LVs did not explain substantially more of the variability of the spatial-temporal gait variables. The quality of the model based on five LVs was good as indicated by Q^2^ = 0.026, R^2^X = 0.22, R^2^Y = 0.61.

PLS-DA classified participants into fallers and non-fallers for ST with AUC = 0.77 (Figure 1A). Figure 1B shows the classification results matrix. In the non-fallers group, the true positive rate and true negative rate of the non-fall group are 84% and 76%, respectively. In the fallers group, the true positive and negative rate, respectively, was 60% and 72%. Note that the AUC for each model is shown in Figure 1A directly, and the true positive rate is directly presented in the diagonal square of the confusion matrix in Figure 1B–D. The true negative rate was also calculated based on the confusion matrix but not directly show in the figures and was presented in Table A1 in the Appendix A.

For DT1 and DT2, the first five LVs were also selected and explained 58.6% and 59.7% variance, respectively. The goodness of the models of DT1 and DT2 was good as indicated by a Q^2^ = 0.018, R^2^X = 0.063, and R^2^Y = 0.18 for DT1, and for DT2 of Q^2^ = 0.054, R^2^X = 0.220, and R^2^Y = 0.6.

Classification of fallers and non-fallers based on DT1 and DT2 gait data obtained an AUC = 0.69 and an AUC = 0.77, respectively. The ROC curve for DT1 and DT2 is shown in Figure 1A. According to the confusion matrix in Figure 1C–D, in terms of DT1, the true positive rate and true negative rate of the non-faller group are 95% and 58%, respectively. While the faller group obtained 17% true positive and 72% true negative rate. For DT2, the true positive rate and specificity of the non-faller group are 88% and 72%, respectively, while the faller group obtained 49% true positive rate and 73% true negative rate.

### 3.2. Identified Gait Variables

Gait variables that contributed most to the classification model were identified by VIP scores with a value >1 (see Figure 2A–C).

For ST, the fallers could be distinguished from non-fallers by higher number of steps, lower mean stride velocity, stride length, and ankle dorsiflexion at heel strike with associated larger mean stance time, stride duration, and ankle plantar flexion at toe-off (for all *p* < 0.05) (Figure 3A).

For DT1 and DT2, fallers were characterized by lower mean stride velocity and stride length, and a lower mean and standard deviation of ankle dorsiflexion at heel strike than non-fallers. On the other hand, non-fallers showed a greater higher number of steps with a larger mean ankle plantar flexion at toe-off than fallers (see Figure 3B). Additionally, for DT2, non-fallers showed a lower mean stance time and stride duration than fallers. The variables from the PLS-DA model with a VIP score >1 were also significantly different between the groups when separately tested (all *p* < 0.05). Figure 3 shows the individual data of the participants in addition to the mean values and confidence intervals of the variables with a VIP score >1 that contribute most to the PLS-DA model during ST and DT1. As can be seen in figure irrespective of the variability between participants within a group, these variables were significantly different between non-fallers and fallers. The results for DT2 were similar to DT1, implying that no major difference was present between walking with an additional cognitive task and walking when performing an additional motor task.

## 4. Discussion

Many people with neurological deficits have an increased fall risk. The present study aimed to develop a model to classify fallers and non-fallers within a heterogeneous group of older adults with neurological disorders. The model is based on spatial-temporal gait variables derived from IMU during walking at a comfortable speed, walking with an additional motor task (DT1), and walking when also performing a cognitive task (DT2) to identify spatial-temporal gait variables that differentiate fallers from non-fallers. We found that gait differed between fallers and non-fallers, and single task walking resulted in the highest classification accuracy in the neurological patients.

### 4.1. Classification Performance of Fallers and Non-Fallers by ST, DT1, and DT2

Overall, the results showed that using PLS-DA fallers could be identified from non-fallers with an AUC of 0.7. Adding DT2, the cognitive dual task, to the model, the AUC was still 0.7, but, with the inclusion of DT1, the motor dual task in the model, AUC decreased to 0.6.

Random forest machine learning method classified Parkinson’s Disease patients versus controls based on gait with an AUC of 0.76 [30]. Given the heterogeneity of the sample in the present study, identification of fallers vs non-fallers with an AUC of 0.7 seems reasonably accurate (Figure 1). The corresponding true positive rate and true negative rate in the fallers group provide more insights into classification performance. In the fallers group, classification based on ST and DT1 produced a similar true negative rate (specificity) around 0.7, suggesting that less than 30% of the non-fallers were classified as fallers.

The true positive rate (sensitivity), however, was lower during DT walking (DT1: 0.17, DT2: 0.49) compared with ST walking (0.72). This finding was unexpected since we anticipated that dual task walking would enhance the differences between fallers and non-fallers. Fallers are expected to have a significant different gait pattern in particularly during DT compared to non-fallers [31] because DT increases the influence of supraspinal control mechanisms on gait compared to ST [32]. The type of DT might be important in this respect. A motor-related DT (walking with a glass of water in hand) improved the discrimination of fallers from non-fallers in otherwise disease-free older adults based on spatial-temporal gait variables [33]. Similarly, a DT with a cognitive component (walking while talking) slowed gait and shortened stride length compared to ST in neurological patients [34]. In the present study, we anticipated that the DT1, with a cognitive component, would demand more cognitive flexibility than DT2, with a motor component. Contrary to this expectation, over 50% of the patients were assigned to the incorrect group. In other words, gait performance under dual task conditions did not improve classification performance compared to gait performance under a single task. Likewise, counting backwards while walking was poorly (high *p*-value > 0.1) associated with falls [35]. The heterogeneity of neurological patients in the present could contribute to the poor classification performance during DT. To illustrate, while falls might be related purely to motor symptoms (bradykinesia, hypokinesia, rigidity) in PD, cognitive dysfunction is likely to contribute to falls in dementia [36]. Therefore, the prediction accuracy of fallers by including DT1 and DT2 in the model would not necessarily improve due to a selective sensitivity of DT1 and DT2 to gait markers of falls. DT1 focus on motor destitution should affect PD, but, for other patients, such as Dementia patients, DT2 with a higher cognitive load presumably has a larger effect on them.

### 4.2. Contribution of Gait Variables to the PLS-DA Classification Model

Gait variables related to the domains of pace (stride velocity), rhythm (stride duration, stance/swing time), variability (standard deviation of ankle dorsiflexion at heel strike), and spatial gait variables (stride length, plantar flexion at toe-off, ankle dorsiflexion at heel strike) contributed significantly to the classification model. These variables appear to be sensitive indicators of gait impairments in a heterogeneous group of neurological patients to identify the risk of falling [37]. In line with previous work [6,23,24], fallers versus non-fallers in the present study walked slower, a different rhythm, higher gait variability, and impaired spatial gait (Figure 3) [38,39]. We extend current data demonstrating that spatial-temporal gait measures can discriminate fallers from non-fallers among healthy older adults by showing that some domains of gait are global and not disease-specific. The classification results in this heterogeneous population might be explained by the fact that multiple types of fallers are included in the dataset and sub-clinical mobility limitations (slow gait, low stability, obesity, arthritis) are randomly distributed among older adults [40]. It is possible that, within different cohorts, they have different risks for falls. The model predicts reasonably well falls because the risks for falls measured through the ‘global’ gait variables are distributed across the cohorts. Therefore, there is a probabilistic chance for a given outcome to predict a fall related to the risk measured with a relatively low error by this outcome. This process is iterated across cohorts, resulting in a relatively accurate fall prediction across cohorts.

### 4.3. Improving Classification Accuracy of a Heterogeneous Population

Nevertheless, there are strategies that can be adopted in the future to improve the current classification model to increase the accuracy of fall classification performance. Fall classification is a multifaceted problem that involves complex interactions between physiological, behavioral, and environmental factors. Most studies that aim to identify falls or classify fallers and non-fallers focus on the factor of motor behavior, such as gait and balance, but do not include other indicators of falls, such as patients’ characteristics [41].

As we know, the combination of intrinsic and extrinsic risk factors contributes to a fall incident. The intrinsic factors include age, fall history, mobility impairments, sleep disturbances, neurological disorders, the presence of co-morbidities, and medication use. Extrinsic factors include slippery surfaces, improper footwear, poor lighting, and clutter [2,42,43]. A comprehensive fall classification should involve the interactions between these risk factors. Clinically, many different test batteries are used that examine gait and balance performance as indicators of fall risk. One of the most well-known and widely-used clinical tests is the Time Up and Go (TUG) [44]. The advantage of TUG is that the test is simple and easy to perform for older adults [45]. Other examples of the clinical test include the Berg Balance Scale (BBS) [46], the Functional Gait Assessment (FGA) [47], and the developed Balance Evaluation Systems Test (BESTest) [48]. However, clinical tests may suffer from ceiling effects, not able to detect relatively small difference, and provide a general score of functioning.

Therefore, more likely, the combined variables from clinical tests and movement measurements could optimize the classification of falls. For example, compared with the fall classification model with only TUG variables, a six-minute walking test equipped with an IMU was added to the TUG to test the model, the classification accuracy of falls in a group of 73 nursing home residents, using a decision tree classifier, increased from 68% to 76% [49].

On the other hand, adding different types of gait variables also could improve the classification performance. In the present study, commonly used spatial-temporal gait variables were calculated from the data collected from wearable sensors during walking, to establish an accurate PLS-DA classification model. However, when combining time/frequency domain and spatial-temporal gait variables together to establish an advanced classification model to discriminate fallers from non-fallers, the accuracy of classification will be enhanced. For example, a Random Forest (RF) classification model classified eleven stroke patients and nine patients with neurological disorders other than stroke (brain concussion, spinal injury, or brain haemorrhage) based on only spatial-temporal gait variables. The classification model performed a moderate testing accuracy of 76.08%. While combining the time domain gait variables and spatial-temporal gait variables and applying a Multilayer Perceptron (MLP) classification model, the classification performance was increased to 84.78% accuracy [21].

The present study results show that in general for fall classification among diverse neurological patients, spatial-temporal gait properties could be used as a biomarker for fallers, irrespective of the specific diagnosis. However, for the identification of pathology specific gait characteristics, e.g., gait features or gait signatures that are unique for a certain diagnostic group (e.g., Parkinson, Stroke, Dementia), other types of gait data, as input to a classification model are needed [22]. The current gait variables are spatial-temporal gait parameters averaged over a number of strides. These parameters do not take into account of time, i.e., fluctuations of walking over a number of strides. Whereas the spatial-temporal gait variables provide more overall information, adding gait variables that include time, the so-called dynamic gait variables, will improve the sensitivity to identify specific diagnostic groups of patients and provide more detailed information for prediction models [50].

Methodologically, gait classification performance can be improved by using a data pre-processing method [51]. Principal Component Analysis (PCA) usually leads to an improvement in classification accuracy [52]. Although PCA is an often-used method for extracting unique features in multidimensional data sets, its assumption (orthogonality) might neglect the significant interactions between the spatial-temporal gait variables. Alternative a less known method for data pre-processing in advance to applying to machine learning methods is the Signature Method. The Signature Method transforms the original data by using path-integral to generate a continuous pathway of the gait data, which results in new feature sets. It not only reduces the redundancy of the data, as well as PCA, but also generates new features that better represent the interactions among these gait variables [53]. The drawback of this new data pre-processing method is that the newly generated features are hard to relate back to the original variables; therefore, it is very difficult to interpret the clinical meanings behind the new features [53].

### 4.4. Selection of Classification Models for Clinical Applications

In order to make a proper program to mitigate fall risks and to improve the gait and mobility impairments, to finally prevent a future fall, advanced computational models, such as machine learning, are anticipated to have the capacity to automatize this process. They can provide a transparent and accurate classification of fallers and non-fallers to assist early identification of fall risks before the actual occurrence of a fall [41].

Different computational methods, such as machine learning, have been used for gait assessment, in general, to construct a model for the classification of different patients and/or age-based groups [50,51]. These computing algorithms should have the capacity to weight the predictive variables, to illustrate the additional clinical value of fall detection, and to assist clinicians in identifying the unique factors that increase falls in a specific population [41].

However, many clinical gait datasets suffer from the co-linear and highly correlated data features, as well as relatively small sample sizes, that are not appropriate for many of these approaches since the accuracy of the widely used machine learning models is dependent on large sets of training data. The more training data, the more accurate, sensitive, and specific the model built. However, sufficient training data may not always be available for the populations in a clinical test. Besides, more variables with less data samples would complicate the model with the low bias high variance to overfit the classification results [51]. For clinical gait analysis, the multiple classical machine learning methods, such as Support Vector Machine (SVM) or Neural Network (NN) approaches, have the advantage to automatically selected the features that are used for a classification model, without any prior feature selection [50]. Yet, for clinical relevance, the results of the computational model parameters are necessary to be translated into meaningful clinical knowledge, despite the complex interactions among the variables leading to the classification [51,54]. The lack of transparency of the construction process in the machine learning classification models limits the reproducibility and clinical interpretation of these advanced computational technologies. This ‘black box’ problem hinders clinical application since clinicians need to understand the specific gait variables for diagnosis [55]. Therefore, in the present study, we applied a method that automatically identified gait parameters, using VIP scores from the PLS-DA model, without a clinical diagnosis as a predictor. PLS-DA is two-fold: Firstly, in clinical gait analysis, many of the gait variables are interrelated. PLS-DA is not impeded by collinearity among variables, which will have a negative impact on the normal Linear Discriminant Analysis. In terms of the fall classification model for clinical gait data, we mostly have more variables than the number of subjects in a dataset. So, one advantage of the PLS-DA model is that we can input various standardized gait variables even more than the number of participants, without prior knowledge to select in advance, but the model still can accurately detect falls. In the present study, we used Leave-One-Out cross-validation (LOOCV) to increase the sample size for training and testing PLS-DA model and to minimize the drawbacks of limited sample size and bias of data [56]. Secondly, PLS-DA is transparent, i.e., several statistical parameters can be derived, such as the VIP scores, to identify the weight of contribution by each variable to the classification model and Q^2^, representing the predictability and validity of the model [57]. In this case, PLS-DA could provide more information to interpret fallers’ gait in clinical. Clinicians using these data science approaches might, at an early stage, improve the identification of patients with fall risk.

When we construct a reliable and accurate computational model, gait patterns can be identified by the model of new patients to identify the at-risk gait on early stage, to diagnose the potential fallers without the prior knowledge of an accurate neurological pathology, and to finally determine the high risk of falling for patients based on their mobility decline [58]. However, human clinical decision-making can be supported and assisted by computational models, such as PLS-DA, but not replace the diagnosis from clinicians. For instance, the identified gait variables could be used for new individuals, to predict the fall risk for potential patients. Moreover, the established computational model might be instrumented in IMU to monitor the interventions in patients’ real-world daily lives and to optimize the efficacy of specific rehabilitation protocols [51].

## 5. Conclusions

The present study classified non-fallers and fallers based on spatial-temporal gait variables derived from IMUs using PLS-DA while walking with or without a DT. The model successfully classified fallers and non-fallers with a satisfactory AUC of 0.69 to 0.77. Thus, differences in gait among neurological patients could be used to identify potential fallers from non-fallers even without DT gait that does not seem to improve classification accuracy among patients with a diverse neurological diagnosis. Number of steps, plantar flexion at toe-off, and ankle dorsiflexion at heel strike, stride length, stride duration, stride velocity, and stance time were sensitive variables to classify fallers and non-fallers. Fallers versus non-fallers have a slow pace and rhythm, high gait variability, and impaired spatial gait pattern. Improving our understanding of how falls risk-related gait impairments in neurological patients could aid the design of tailored fall-prevention interventions to decrease the fall risk for people with neurological deficits.

## Figures and Tables

**Figure 1 sensors-20-04098-f001:**
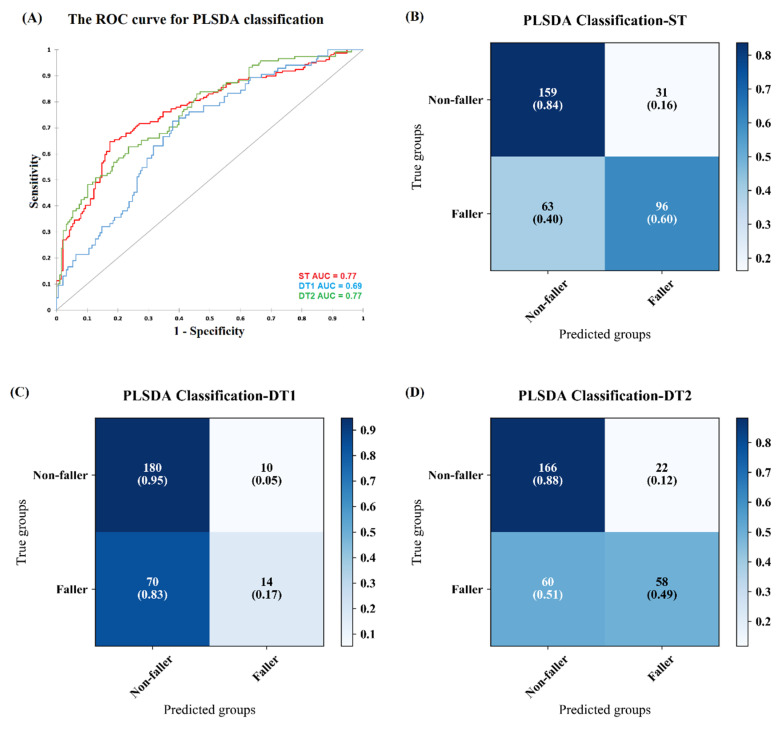
(**A**) shows the receiver operating characteristic (ROC) curves for partial least square discriminant analysis (PLS-DA) classification, based on ST (yellow), DT1 (green) and DT2 (blue) gait variables. (**B**–**D**) shows the classification matrix. The x-axis represents the participants in the predicted groups and the y-axis shows the participants in the original groups. The dark blue means more participants were assigned in this group. The numbers of participants and the percentages they occurred in the original group are shown in the squares and braces. DT1 = walking and checking boxes on a paper sheet; DT2 = serial 7 s subtraction.

**Figure 2 sensors-20-04098-f002:**
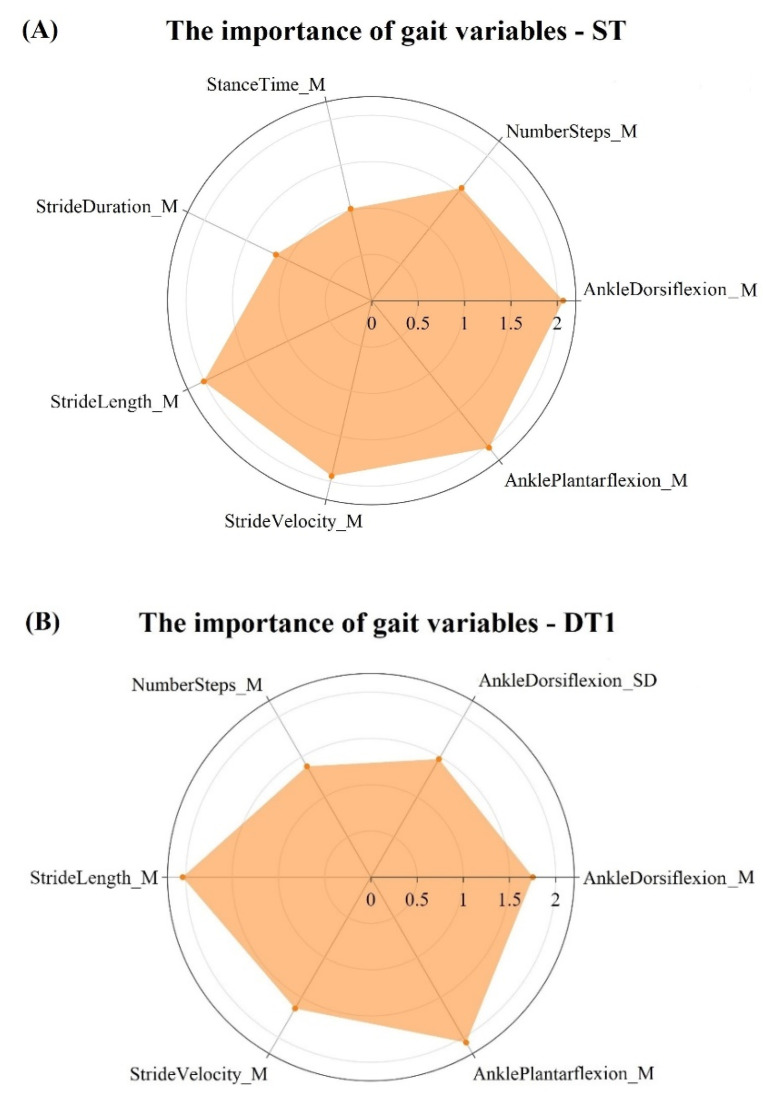
(**A**–**C**) shows the importance of the gait parameters by orange area (VIP > 1) from ST, DT1, and DT2. M = mean, SD = standard deviation. DT1 = walking and checking boxes on a paper sheet; DT2 = serial 7 s subtraction.

**Figure 3 sensors-20-04098-f003:**
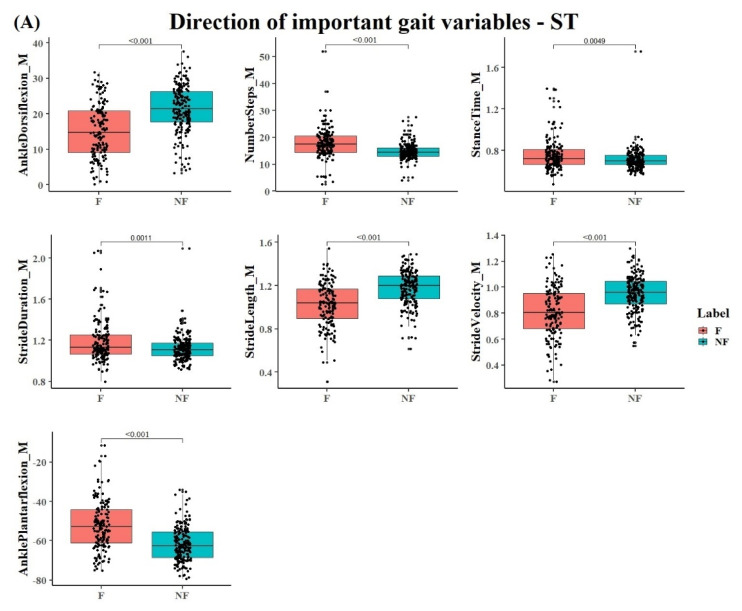
(**A**,**B**) show the direction of variables that contribute more to the PLS-DA model in and DT1. The x-axis represents the groups of fallers and non-fallers, and the y-axis shows the coefficients of each variable in each square. The vertical bars indicate the confidence interval. Dots show the individual data of the participants. DT1 = walking and checking boxes on a paper sheet. Note that the results for DT2 were similar as for DT1.

**Table 1 sensors-20-04098-t001:** Demographics of participants for the single task (ST) a motor dual tasks (DT1) and a cognitive dual task (DT2).

Tasks	Non-Fallers		Fallers		
	ST and DT1	DT2	ST	DT1	DT2
No. Males	115	115	88	41	64
No. Females	75	73	71	43	54
No. Total	190	188	159	84	118
Age, years	61.6 ± 12.2	61.5 ± 12.2	65.0 ± 12.7	61.8 ± 12.5	65.0 ± 12.5
Height, m	1.73 ± 0.1	1.73 ± 0.1	1.70 ± 0.1	1.71 ± 0.1	1.72 ± 0.1
Weight, kg	82.04 ± 16.25	82.04 ± 16.2	76.31 ± 14.87	75.97 ± 15.56	77.07 ± 14.61
BMI, kg/m^2^	27.22 ± 4.79	27.25 ± 4.8	26.08 ± 4.34	25.8 ± 4.33	26.02 ± 3.97

Values are mean ± SD, BMI = body mass index, ST = walking at a comfortable speed without an additional task, DT1 = walking and checking boxes on a paper sheet, DT2 = serial 7 s subtraction.

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
