# Peer review of "Classification of Neurological Patients to Identify Fallers Based on Spatial-Temporal Gait Characteristics Measured by a Wearable Device"

_sensors, 2020, doi:10.3390/s20154098_

Round 1

Reviewer 1 Report

This study aims to propose a quantitative method to classify fallers and not-fallers in a heterogeneous population of neurological patients by analysing spatial-temporal gait parameters. A further purpose of the study is to identify the gait characteristics mainly impacting on this classification model. The authors enrolled 384 patients suffering from several neurological disorders and classified them clinically as fallers and non-fallers based on the history of falls within the last two years. The participants of the study were asked to walk during a single task, a dual-cognitive task and a dual-motor task while monitored by an inertial sensor system. The proposed model using a partial least square discriminant analysis showed a rather good performance in distinguishing fallers from non-fallers, especially during the single walking task. Moreover, some gait variables that contributed most to the classification model were also identified. The authors concluded that the findings of this study could help in improving fall-prevention strategies in patients with neurological disorders.

The study is of interest and the manuscript is written rather well.  However, I have several comments possibly helping to improve the overall scientific quality of the paper:

  • The introduction could be improved by expanding the background concerning possible mechanisms leading to falls. Several recent reviews addressed this issue also examining the role of wireless sensors in neurological disorders.
  • Previous experimental studies addressed the topic of the objective classification of fallers (e.g. Howcroft et al. 2016, 2017; Drover et al. 2017; Liang et al. 2015). Accordingly, the authors should improve the background of their study by considering previous findings and clarifying the novelty of the proposed approach.
  • It is not clear whether the fallers group also included patients with a history of incidental falls possibly depending only on extrinsic risk factors. This issue would negatively impact on the performance of the proposed model.
  • To properly interpret the performance of the proposed model, it would be necessary to know the composition of the fallers and non-fallers groups (e.g. how many parkinsonian or epileptic patients in the two groups?). The authors could clarify this point by reporting this information in one table.
  • A table reporting sensitivity, specificity, accuracy, positive and negative predictive values would help the reader to easily understand the results and the performance of the proposed model.
  • It would be interesting to assess the performance of the proposed classifier based on the number of falls reported in the patients' history.
  • The lack of clinical measures of gait performance in the two groups (e.g. TUG times) partially limits the interpretation of data. Indeed, it is not possible to understand the severity of the gait dysfunction in the two groups. The authors should discuss this issue in a limitation paragraph.

Minor points:

  • The 4.3 and 4.4 paragraphs are too verbose and could be summarised. Also, there are some repetitions (e.g. Pages 9-10; Lines 238-239, 293-294).
  • The expression “postural control of gait” (Page 8; Lines 219-220, 224) could be confusing since this study did not consider sensor-based measures of balance or postural control.

Author Response

Dear Reviewer,

We provide a point-by-point response to your comments, please kindly refer to the attachment.

Best regards,

Yuhan, on behalf of all authors

Reviewer 2 Report

The paper describes an interesting approach for the analysis of functional motor variables (both in time and space domains) to classify faller and non-fallers.

The methodology is well presented and the discussion section is deep and structured.

Results need improvement. According to the different categories of subjects or tests a set of weighting coefficients could be included to prioritize some variable with respect to others. This is not considered as possible solution. Please discuss.

Author Response

(The authors gave the same response as above.)

Reviewer 3 Report

This paper presented the prediction of fall status from spatial-temporal gait characteristics measured by a wearable device in a heterogeneous population of neurological patients. The least square discriminant analysis (PLS-DA) was applied to classify fallers and non-fallers, and the results of three gait tasks are analyzed and discussed. The topic of this paper is within the scope of the journal; however, the outcomes are unconvincing. Therefore, before the paper can be published, the following issues must be addressed to improve the quality of this paper:

(1) The gait recognition/prediction is a very mature research area with plenty of effective methods and algorithms that have been applied in practical usage. Therefore, whether other type of methods (for example, the methods listed and discussed in section 4.4) can achieve better fall status prediction results than the PLS-DA method need to be carefully discussed via the experimental comparison. In addition, the comparison between PLS-DA and other statistical analysis methods such as PCA/HCE/PLS/OPLS, etc., also need to be conducted.

(2) To justify the effectiveness of the proposed method, the influence of the number of samples (participants) on the prediction accuracy of the model needs to be discussed. As the AUC of three gait tasks only achieved 0.6~0.7, from the perspective of the reviewer, the current prediction accuracy cannot satisfy the practical clinical needs. Whether the AUC will increase or decrease along with the variance of samples should be studied carefully.

(3) Some figures should be enclosed in the paper to demonstrate how the IMU is installed/aligned on the ankle joint of subjects.

Author Response

(The authors gave the same response as above.)

Round 2

Reviewer 1 Report

The authors properly addressed my comments. 

Reviewer 3 Report

The response letter explains the relevant issues well to the reviewer. It is suggested that some replies can be briefly reflected in the manuscript to help the readers better understand the paper.